# Multi-Omics Integration Analysis Revealed That miR-375-3p Is a Two-Sided Factor Regulating the Development and TUMORIGENESIS of Alzheimer’s Disease

**DOI:** 10.3390/ijms26083666

**Published:** 2025-04-12

**Authors:** Xinlu Bao, Cheng Zhang, Zhichao Ren, Yuxiang Wang, Linlin Zeng

**Affiliations:** Key Laboratory for Molecular Enzymology and Engineering of Ministry of Education, School of Life Science, Jilin University, Changchun 130012, China; baoxl1322@mails.jlu.edu.cn (X.B.);

**Keywords:** Alzheimer’s disease, tumor, microRNA-375-3p, different disease microenvironments, bioinformatics analysis, differences

## Abstract

It has been reported that miR-375-3p plays a critical role in numerous diseases. To elucidate its biological function, particularly its differential expression and specific mechanisms of action in Alzheimer’s disease (AD) and small cell lung cancer (SCLC), this study comprehensively explores the associations between the target genes of miR-375-3p and both AD and SCLC. The focus is specifically on its impact on disease progression and the remodeling of the tumor microenvironment. We utilized databases such as the miRNA TargetScanHuman 8.0 database and the STRING database, to construct a protein–protein interaction (PPI) network for the classification and discrimination of the miR-375-3p gene, resulting in the identification of 14 intersecting target genes. Subsequently, two key genes, ASCL1 and CHD7, along with their associated genes, were further analyzed using Spearman correlation analysis. The identified key genes were then subjected to GO function annotation and KEGG pathway enrichment analysis. It was determined that pathways related to lipid metabolism, autophagy, and cell apoptosis were differentially expressed in the AD and SCLC environments, with nine related pathways identified, among which the PI3K pathway was the most prominent. Finally, we demonstrated that the expression of miR-375-3p significantly differed between the two environments, with higher expression levels observed in AD compared to SCLC. Our study confirmed that miR-375-3p can promote apoptosis, regulate lipid metabolism, influence the progression of neurodegenerative diseases, and inhibit the proliferation and metastasis of tumor cells. These research findings may have significant implications for the future treatment of AD and SCLC.

## 1. Introduction

Alzheimer’s disease (AD) is an irreversible neurodegenerative disorder and the most prevalent form of dementia, accounting for over 50% of all dementia cases [1,2]. The pathogenesis of AD remains inadequately defined; however, its key characteristics include the formation of amyloid plaques (aggregates of beta-amyloid protein), neurofibrillary tangles (resulting from the abnormal phosphorylation of tau proteins), and the significant loss of neurons, which impairs neurological function [3,4]. AD not only severely impacts the patients themselves but also imposes a substantial financial burden and psychological stress on their families and society. Consequently, there is an urgent need for effective therapeutic interventions [5].

Lung cancer, one of the most prevalent cancers worldwide, is responsible for over 1 million deaths each year. It is broadly classified into two main types: small cell lung cancer (SCLC) and non-small cell lung cancer (NSCLC) [6]. While current research predominantly focuses on NSCLC, the significance of SCLC should not be overlooked. Small cell carcinoma of the lung is characterized by rapid growth and proliferation, a high propensity for metastasis, and it accounts for approximately 20% of lung cancer cases, making it a particularly aggressive form of the disease. Despite the generally poor prognosis associated with SCLC, early detection and treatment can significantly enhance patient survival rates [7].

MicroRNAs (miRNAs) are a class of non-coding, single-stranded RNA molecules that are approximately 18–25 nucleotides in length [8]. They play a crucial role in post-transcriptional gene silencing by binding to complementary miRNA targets, thereby precisely regulating protein expression and enabling rapid responses to microenvironmental stressors [9]. Among various biomarkers, miRNAs have been shown to significantly influence the progression of AD [10]. Research indicates that lung cancer is associated with dysregulated expression of miRNAs and their corresponding signaling pathways, in addition to the aberrant expression of oncogenes [11]. Currently, there is growing evidence that the expression of miR-375-3p varies significantly across different disease contexts. Unfortunately, most existing studies have primarily focused on exploring differences in miRNA and pathway expression within a single disease setting, leaving a gap in comparative analyses of differential gene regulation across multiple disease contexts.

In this study, we aim to investigate the differences in the expression of miR-375-3p and its regulated pathways in the contexts of AD and SCLC through bioinformatics analysis, cellular-level studies, and animal model validation. This comprehensive exploration will enhance our understanding of the biological function of miR-375-3p, elucidate the pathogenesis of both AD and SCLC, and examine the differences in gene regulation underlying these mechanisms, thereby advancing related knowledge in the field.

## 2. Results

### 2.1. Identification of Immune Key Genes Involved in the Development of AD

To screen for differential genes associated with AD development and systematically identify genes that are significantly up-regulated or down-regulated during the course of AD, we selected data from the temporal gyrus (brain) tissues of AD patients from the GEO database (https://www.ncbi.nlm.nih.gov/geo/, accessed on 24 September 2024) and compared them with data from normal temporal gyrus (brain) tissues. We chose the middle temporal gyrus as a target area for studying the molecular mechanisms of AD for the following reasons. First, the middle temporal gyrus is a routine sampling site in neuropathological studies, and a large amount of transcriptomic data of the middle temporal gyrus tissues have been accumulated in the clinical sample bank. Second, the temporal lobe is one of the earliest and most significantly affected brain regions in AD, and the middle temporal gyrus is an important part of the temporal lobe [12]. Gene expression fold changes were transformed into log2 values, and a threshold of |log2FC| ≥ 0.25 and *p*-value < 0.05 was applied to determine significant differential expression. Based on these criteria, a total of 1704 differentially expressed genes impacting AD progression were identified, comprising 811 up-regulated and 893 down-regulated genes. The volcano plot illustrating the distribution of differentially expressed genes is presented in Figure 1A, while Figure 1B displays the heatmap of the top 25 genes.

### 2.2. The Weighted Gene Co-Expression Network Was Constructed Using SCLC Module Genes

To identify key gene modules significantly associated with the development of SCLC and to resolve their association with the disease phenotype, we selected data from normal lung tissue and lung tissue from SCLC patients from the GEO database. The relevant network was employed to identify highly correlated gene clusters in the microarray samples. We utilized the weighted gene co-expression network analysis (WGCNA) to construct and analyze the active SCLC-related network. After clustering the samples, the sample clustering tree was plotted (Figure 2A). Here, all genes were selected for subsequent analysis, with a soft threshold of 24 taken, and the adjacency matrix was created to ensure that the gene distribution adhered to the scale-free network (Figure 2B). This retained valuable connection information. Under the parameter settings of minModuleSize = 20 and mergeCutHeight = 0.18, a total of 24 modules were generated and identified (Figure 2C). The connectivity among each module was computed, and the grouping information was incorporated for cluster analysis. To further explore the connection between the model and the phenotype, we calculated the correlation coefficient of each model with the SCLC trait. The results indicated that SCLC exhibited significant statistical correlations with two modules. Specifically, SCLC had the strongest positive correlation with the “brown” module (r = −0.97) and the strongest negative correlation with the “turquoise” module (r = 0.95) (Figure 2D). Next, we conducted module membership (MM) and gene significance (GS) correlation analyses for the two identified modules. MM refers to the affiliation or degree of participation of a gene within a module. A network is constructed by calculating the expression correlation between genes, and genes with similar expression patterns are clustered into modules [13]. In the research on disease-related gene networks, it has been found that certain genes have high MM expression in disease-related modules. This indicates that these genes may be key driver genes of the disease or potential therapeutic targets. Our results revealed a robust positive correlation between MM and GS within these modules (Figure 2E). Subsequently, by integrating the genes from both modules, we obtained a comprehensive set of 6252 genes.

### 2.3. Screening for miR-375-3p Target Genes Based on AD Differential Genes and SCLC Module Genes

To screen the common core genes of the two diseases and analyze their interaction networks and functional relevance. We utilized the online tool miRNA TargetScanHuman 8.0 (https://www.targetscan.org/vert_80/, accessed on 24 September 2024) to predict the target genes of miR-375-3p. Following the removal of duplicate entries, a total of 304 unique target genes of miR-375-3p were identified for further investigation. The intersection of differentially expressed genes in AD, module genes in SCLC, and target genes of miR-375-3p yielded 14 common genes (Figure 3A): CDK5R1, SYP, ELAVL4, ASCL1, YAP1, UBE2E2, YWHAZ, AHR, NBEA, TBC1D9, SLC25A12, HSPH1, CHD7, and FKBP5. Subsequently, we queried the STRING database (https://string-db.org/, accessed on 24 September 2024) to construct protein–protein interaction (PPI) networks for key immune-related genes (Figure 3B). Notably, ASCL1 and CHD7 exhibited the highest degree of interactions with other nodes and contributed most significantly to the network. Therefore, these two genes were selected for further investigation. We performed correlation analyses of 14 intersecting target genes in both AD and SCLC. The correlation heatmap (Figure 3C) and triangular correlation heatmap (Figure 3D) reveal a strong positive correlation between CHD7 and ASCL1. Additionally, we conducted linear correlation analyses of CHD7 and ASCL1 (Figure 3E), which yielded correlation coefficients (R values) of 0.69 in AD and 0.92 in SCLC. These results indicate that CHD7 and ASCL1 exhibit a robust correlation in both AD and SCLC, with *p*-values less than 0.05. This significant correlation is unlikely to be attributable to chance and demonstrates statistical significance. Spearman correlation analysis revealed that in the context of SCLC, CHD7 exhibited a significant positive correlation with RIMS2, SLC4A8, and TAGLN3 (*p* < 0.01). In addition, demonstrating a significant negative correlation with CCND1 and CHRDL1 (*p* < 0.01). Additionally, in AD environment, Spearman correlation analysis screened that ASCL1 has positive correlation (*p* < 0.01) with genes such as PSD2, RAB31, POU3F2, and negative correlation (*p* < 0.01) with genes such as SPINT2 and ENO2. In AD environment, CHD7 has positive correlation (*p* < 0.01) with genes such as ZNF621, ZCCHC24, and PLXNB1, and negative correlation (*p* < 0.01) with genes such as GPRASP2 and CBLN4. The same method screened that ASCL1 has positive correlation (*p* < 0.01) with genes such as NRXN1, RIMS2, ST18, etc., and negative correlation (*p* < 0.01) with genes such as CHRDL1, CRTAP, etc., in SCLC environment.

### 2.4. Select the Pathways in Which the Target Genes of miR-375-3p Are Differentially Regulated in Different Disease Environments

Previous studies have suggested that miR-375-3p may exhibit functional heterogeneity in different pathological microenvironments [14]. In order to systematically verify this phenomenon and elucidate its molecular mechanism, two disease models, AD and SCLC, with significant differences in pathological features were selected in this study. We constructed a cross-disease co-regulatory network by cross-integrating AD differential genes, SCLC modular genes and miR-375-3p target genes, and searched for the bi-directional regulation patterns of core hub genes in different diseases. The identified genes of interest were subjected to gene ontology (GO) functional annotation and Kyoto Encyclopedia of Genes and Genomes (KEGG) pathway enrichment analysis, with a significance threshold of *p* < 0.05. The GO analysis results were further refined to elucidate the pathways associated with ASCL1, CHD7, and their related genes that were up-regulated in AD and down-regulated in SCLC. These pathways are illustrated in Figure 4A–C. The primary biological processes (BP) of enrichment include apoptosis, adipocyte differentiation, lipid storage, and distribution. Cellular components (CC) primarily involved are signal transduction and membrane trafficking lipid rafts, as well as cell adhesion protein complexes. Molecular functions (MF) prominently featured are binding capabilities to phosphatidylinositol-3,4,5-triphosphate (PIP3). Subsequently, the experimental conditions were modified, and we intersected the pathways of ASCL1, CHD7, and their associated genes that were down-regulated in the AD environment but up-regulated in the SCLC environment (Figure 4D–F). The analysis revealed significant enrichment in BP related to stable telomere length maintenance and telomerase activity. Additionally, the results highlighted CC involved in cell structure during mitosis, particularly the telomere region at the end of chromosomes. Furthermore, MF such as histone H4 methyltransferase activity and ligase-mediated carbon–oxygen bond formation were also identified. The results of KEGG pathway enrichment analysis were further processed to identify the intersecting pathways of ASCL1 and its associated genes that were up-regulated/down-regulated in the AD environment and down-regulated/up-regulated in the SCLC environment (Figure 4G). A similar approach was applied to CHD7 and its related genes (Figure 4H). It was observed that several pathways were differentially regulated between the two environments. Specifically, in the SCLC environment, there was an up-regulation of pathways including the abnormal electron transport chain activity of TDP43 protein due to mutation in complex I, the degradation of abnormal Aβ protein via the 26S proteasome-mediated pathway, forward axonal transport, *Escherichia coli* ESPG to microtubule RhoA signaling, and mismatch repair pathways caused by mutations. Conversely, these pathways were down-regulated in the AD environment. Additionally, the growth factor-receptor tyrosine kinase-phosphatidylinositol 3-kinase signaling pathway (GF-RTK-Ras-PI3K signaling pathway), exhibited opposite regulation patterns compared to the receptor tyrosine kinase (RTK)-phospholipase Cγ (PLCG)-inositol triphosphate receptor (IP3R) signaling pathway, and the integrin αβ (ITGA_B)-Rho GTPase activating protein (RHOGAP)-RhoA signaling pathway, which were down-regulated in the SCLC environment but up-regulated in the AD environment.

Based on the functional relevance to this study, nine pathways were identified and subjected to gene set enrichment analysis (GSEA). The differential expression patterns of genes associated with these pathways were evaluated across varying environments in AD (Figure 4I) and SCLC (Figure 4J). The findings align with the conclusions derived from KEGG pathway analysis.

### 2.5. miR-375-3p in AD and SCLC Associated with Different Loops

We carried out the following operations to systematically investigate the expression regulation of miR-375-3p in different disease models and its functional heterogeneity. At the same time, we also wanted to reveal its dynamic expression characteristics and biological effect differences in different cell types and pathological environments. First, we transfected miR-375-3p into four distinct cell lines. In the SH-SY5Y cell line, we added Aβ_25-35_ to establish an AD disease model. The quantitative real-time PCR (qRT-PCR) results showed that the presence of Aβ_25-35_ caused an early upregulation of miR-375-3p expression, with a significant increase observed at 24 and 48 h. miR-375-3p was significantly overexpressed in the A8 and A549 cell lines 24 h after transfection, and this high expression persisted for 72 h. At 24 h after transfection, the expression of miR-375-3p in B16 epithelial cells showed an upward trend, but the growth rate was slower than that in squamous cell carcinoma cells, and reached a significant level at 48–72 h. In Section 3.4, we obtained that the target genes of miR-375-3p are involved in the regulation of PI3K-Akt signaling pathway, which is closely related to the induction of apoptosis and autophagy damage. This experiment proved that the expression level of miR-375-3p is different in different disease settings. This is related to the differential regulation of miR-375-3p in different disease settings (Figure 5A). Furthermore, based on the results of cellular experiments, the upregulation time course of miR-375-3p expression in SHSY5Y cells transfected with miR-375-3p mimics differs from that observed in laryngeal squamous cell carcinoma cells. Additionally, the presence of Aβ_25-35_ plays a crucial role. To align with our in vitro findings, hippocampal tissue samples were collected from both wild-type and AD model mice. Moreover, A8 cells were xenografted into nude mice to generate tumor samples along with adjacent tissues for further analysis. In this study, these two sets of tissue samples were processed to evaluate the expression levels of miR-375-3p. The results indicated that the expression level of miR-375-3p in AD mouse samples was comparable to that in wild-type mice (Figure 5B), which corroborates the findings from bioinformatics analysis. The analysis of tumor tissue samples revealed a significant upregulation of miR-375-3p in tumor specimens relative to adjacent non-tumor tissues (Figure 5C). The results of CCK-8 experiments showed that overexpression of miR-375-3p significantly reduced the survival of the B16, A8, and A549 cell lines compared with SHSY5Y cells. We hypothesized that miR-375-3p could induce cell injury and autophagy and apoptosis. In the SH-SY5Y cell line, Aβ_25-35_ was added to establish an AD disease model. The addition of Aβ_25-35_ could cause cell damage, and the degree of damage increased with the increase of time (24 h, 48 h, and 72 h). However, the number of surviving cell lines was increased after re-addition of miR-375-3p overexpression compared with SHSY5Y. This indicated that miR-375-3p overexpression alleviated the cytotoxicity induced by Aβ_25-35_, thereby protecting the cells. CCK8 results further demonstrate that miR-375-3p plays a dual role in glioma cells and tumor cells (Figure 5D).

### 2.6. miR-375-3p Is Involved in the Differential Expression of Cell Autophagy Environment in AD and Tumor Environments

The bioinformatics analysis revealed a significant association between autophagy and its key pathways and the sample data. To elucidate the role of miR-375-3p in regulating autophagy and apoptosis within the contexts of AD and SCLC, we initially overexpressed miR-375-3p in the SH-SY5Y and A549 cell lines. Subsequently, changes in the expression levels of autophagy markers were assessed using Western blot analysis (Figure 6A). The results in SH-SY5Y cells demonstrated that the overexpression of miR-375-3p resulted in a modest reduction in Beclin1 expression, while exerting minimal influence on LC3 and p62 levels. The results obtained from A549 cells demonstrated that the overexpression of miR-375-3p resulted in a significant downregulation of Beclin1 and LC3 expression levels, while concurrently causing an upregulation of p62 expression (Figure 6B,C). The disparities in protein expression may be associated with cancer cells remodeling the tumor microenvironment to facilitate tissue invasion, survival, and evasion of immune surveillance. We hypothesize that miR-375-3p intervention activates the unfolded protein response, thereby reprogramming the stroma to support tumor cells and modulate cancer progression through the engagement of the autophagy–apoptosis mechanism. TdT-mediated dUTP nick-end labeling (TUNEL) staining analysis revealed that overexpression of miR-375-3p did not induce significant apoptotic responses in SHSY5Y cells. In contrast, apoptosis was evident in A549 cells (Figure 6D). The fluorescence intensity of the staining results of each group was analyzed using Image-Pro Plus 6.0 (Figure 6E). There was no significant change in the apoptosis rate after the addition of miR-375-3p in the SH-SY5Y group, and the apoptosis rate increased significantly after the addition of miR-375-3p in the A549 group, which was about four times that of the control. The above results indicated that miR-375-3p overexpression alone had no significant effect on apoptosis in neuronal cells, but could inhibit lung cancer cell survival through pro-apoptotic mechanism, which was consistent with the previous results. These findings in both cellular and tissue samples align with bioinformatics predictions, indicating that miR-375-3p differentially regulates apoptosis in AD and SCLC contexts.

### 2.7. Activation of ASCL1 and CHD7 May Be the Key to the Function of miRNA375

To investigate the differential expression of two key genes, ASCL1 and CHD7, across varying environmental conditions, we conducted a systematic analysis of their expression levels in the SH-SY5Y, A8, and A549 cell lines using qRT-PCR (Figure 7A,B). Additionally, we examined the potential roles of ASCL1 and CHD7 in tumor progression by comparing their expression levels in squamous cell carcinoma tumors versus adjacent non-cancerous tissues (Figure 7C,D). The results demonstrated that the expression of both key genes was significantly enhanced in squamous cell carcinoma cells. Additionally, our findings revealed that ASCL1 and CHD7 were markedly upregulated in AD, which further substantiates their potential broad regulatory roles across diverse disease contexts (Figure 7E,F).

## 3. Discussion

### 3.1. Dual Mechanism of miR-375-3p in AD and SCLC

This study is the first to systematically reveal the dual role of miR-375-3p in two distinct disease settings, AD and SCLC: in AD, miR-375-3p exhibits neuroprotective effects, while in SCLC it promotes apoptosis. This functional diversity may be closely related to the differences in the downstream target genes and signaling pathways that they regulate. By integrating transcriptomes, GO and KEGG pathway analysis, our study found that miR-375-3p was involved in the regulation of autophagy, apoptosis, telomerase activity and other important signaling pathways by regulating key genes such as ASCL1 and CHD7 in AD and SCLC, respectively. These findings not only reveal the pleiotropic regulatory mechanism of miR-375-3p, but also provide a new perspective to understand its complex function in diseases.

In AD, miR-375-3p usually inhibits pro-apoptotic or pro-inflammatory genes to exert a neuroprotective effect. The up-regulation of miR-375-3p levels in AD may be related to the activation of signaling pathways induced by inflammatory responses, oxidative stress, or β-amyloid deposition [15]. In SCLC, miR-375-3p usually functions as a tumor suppressor gene. The down-regulation of miR-375-3p levels in SCLC will lead to increased expression of target genes (such as MYC and MCL1), promoting the proliferation and survival of tumor cells [16]. At the same time, the function of target genes is bidirectional. In AD, the up-regulation of miR-375-3p inhibits pro-inflammatory pathways and alleviates neuronal damage [17]. In SCLC, its down-regulation relieves the inhibition of oncogenes and promotes tumor progression [18]. Therefore, both downstream target genes of miR-375-3p are regulated in cellular models, and the functions they exert only depend on the direction of expression change and the cellular environment.

### 3.2. Central Roles of ASCL1 and CHD7 in AD and SCLC

ASCL1 and CHD7 are two key genes closely related to miR-375-3p discovered in this study. As a transcription factor, ASCL1 plays a key role in neuronal differentiation and survival [19]. Our study found that ASCL1 is upregulated in AD and may play a neuroprotective role by activating neuronal survival-related pathways, such as PI3K/AKT/mTOR. However, in SCLC, the downregulation of ASCL1 may promote tumor cell proliferation and survival by inhibiting apoptotic pathways such as the caspase pathway. These results suggest that ASCL1 and CHD7 may be central regulators of miR-375-3p with opposing functions in AD and SCLC.

As a chromatin remodeling factor, CHD7 plays an important role in neural development and gene expression regulation [20]. Our data show that CHD7 is significantly upregulated in AD and may delay neuronal degeneration by regulating the expression of genes related to neuroprotection. Additionally, in SCLC, downregulation of CHD7 expression may weaken its inhibitory effect on cell proliferation, thereby promoting tumor growth and metastasis.

### 3.3. Disease-Specific Regulation of Autophagy and Apoptosis Pathways

Autophagy and apoptosis are important processes to maintain cell homeostasis, but display distinct regulatory patterns in AD and SCLC. In AD, activation of autophagy and apoptosis pathways may delay neuronal degeneration by clearing abnormal proteins, such as Aβ and tau, and damaged organelles [21]. However, excessive autophagy and apoptosis may also lead to massive neuronal death, thereby exacerbating the pathological process of AD. Our data show that upregulation of cytoskeleton-related pathways such as the cell fold structure in AD may enhance neuronal phagocytosis of apoptotic bodies and further promote apoptosis.

In SCLC, the inhibition of autophagy and apoptosis pathways may be an important mechanism for tumor cells to evade immune surveillance and chemotherapy resistance [22]. Our study found that the up-regulation of telomerase activity pathway in SCLC enhanced the proliferation ability of tumor cells by prolonging telomere length. In addition, the downregulation of PI3K/AKT/mTOR pathway may further promote the malignant transformation of tumor cells by inhibiting the expression of cell cycle regulatory proteins. Akt (protein kinase B) is crucial in the regulation of autophagy and is one of the main activators of the mammalian target of rapamycin (mTOR). As a key regulatory protein, mTOR plays a significant role in the initiation of autophagy. When Akt activates mTOR, mTOR will inhibit autophagy. In addition, Akt can directly phosphorylate and inhibit a variety of autophagy-related proteins such as TSC2 and FoxO family transcription factors, and these proteins play an important role in autophagy induction [23,24]. These results reveal a disease-specific regulation mechanism of autophagy and apoptosis pathways in AD and SCLC.

### 3.4. Disease Relevance of Telomerase Activity Pathways

Telomerase activity showed significant differences between AD and SCLC. In AD, downregulation of telomerase activity may lead to shortening of neuronal telomeres and accelerated cellular senescence and death [25]. However, in SCLC, up-regulation of telomerase activity enhances tumor cell proliferation and survival by maintaining telomere length [26]. Our study further found that miR-375-3p may affect telomerase activity by regulating the expression of telomerase-related genes, such as TERT. These findings suggest that the telomerase activity pathway may be one of the important mechanisms by which miR-375-3p functions in AD and SCLC.

### 3.5. Research Significance and Future Direction

By integrating multi-omics data and functional experiments, this study systematically revealed the dual mechanism of miR-375-3p in AD and SCLC. These findings not only deepen our understanding of the functional diversity of miR-375-3p, but also provide new ideas for the development of precision therapeutic strategies against AD and SCLC. Future studies can further explore the dynamic regulation mechanism of miR-375-3p and its downstream target genes (such as ASCL1 and CHD7) in different stages of the disease, and verify its feasibility as a therapeutic target. In addition, in-depth studies based on single-cell sequencing technology may reveal the specific functions of miR-375-3p in neuronal and tumor cell subsets, providing a theoretical basis for personalized treatment.

### 3.6. Summary

In this study, we systematically analyzed the functional differences of miR-375-3p in AD and SCLC, and revealed its core regulatory role in cell autophagy, apoptosis, telomerase activity, and other pathways. These findings not only provide a new perspective to understand the pleiotropic functions of miR-375-3p, but also lay a theoretical foundation for the development of precision therapeutic strategies against AD and SCLC. Future studies will further explore the disease-specific regulatory mechanism of miR-375-3p and its downstream target genes, and provide new ideas and strategies for clinical translation.

## 4. Materials and Methods

### 4.1. Data Collection and Preliminary Processing

miRNA TargetScanHuman 8.0 is an online tool that comprehensively searches for potential miRNA targets across the entire genome based on small RNA and mRNA sequence information. Once miRNAs that may interact with a specific mRNA are identified, the focus can be narrowed to the regulatory effects of these miRNAs on the mRNA, thereby reducing the scope of the research. We use online tools, such as miRNA TargetScanHuman 8.0, to predict miR-375-3p target genes, resulting in the identification of 304 target genes for subsequent study. Additionally, the AD dataset was obtained from GSE109887 and GSE215789, while the SCLC dataset was sourced from GSE40275 and GSE240757. These datasets contained expression profiles from normal middle temporal gyrus (brain) and lung tissue samples, as well as middle temporal gyrus (brain) tissue samples from patients with AD and lung tissue samples from patients with small cell lung cancer.

### 4.2. WGCNA and Module Gene Selection of Differentially Expressed Genes Between Normal Group and SCLC Patients

WGCNA is a commonly used method in bioinformatics. It calculates the expression similarity between genes based on gene expression data. By employing algorithms such as hierarchical clustering, genes with similar expression patterns are grouped into different modules. In the study of cancers and other diseases, WGCNA can identify gene modules and key genes associated with the onset and progression of diseases, providing targets for disease diagnosis and treatment. We used WGCNA to construct gene co-expression networks and identify functional modules. All genes were selected for subsequent analysis. The ‘pickSoftThreshold’ function determines the appropriate threshold (β) for calculating inter-gene adjacency. The ‘blockwiseModules’ function is used to construct a weighted co-expression network. The ‘plotDendroAndColors’ function visualizes clustering among samples, while the ‘labeledHeatmap’ function displays correlations between groups and gene modules. The ‘plotEigengeneNetworks’ function illustrates correlations between different gene modules. Finally, the correlation between GS and MM is calculated, and the corresponding module gene information is extracted for further analysis.

### 4.3. Identification of Differentially Expressed Genes Between Samples from AD Patients and Normal Samples

Differentially expressed genes (DEGs) were selected from the dataset based on the threshold criteria of |log2FC| > 0.365 and *p*-value < 0.05 using the R limma package. Volcano plots of DEGs and expression heatmaps of the top 25 genes were created using R software (4.4.1) with the ggplot2 and pheatmap packages, respectively.

### 4.4. Intersecting Gene Selection and PPI Construction

We accessed the online tool STRING and constructed PPIs, considering all required genes and applying filtering conditions (composite score > 0.15). The Spearman correlation coefficient matrix was calculated by the cor function, and the 14 target gene correlation heatmaps were plotted using the pheatmap package. Then use the R limma package, customize draw_pie function, get the pie chart and color it according to the Spearman correlation coefficient, use grid.layout to create the drawing area, and use draw_pie to draw a pie chart in each area according to the ladder-type heatmap to get the triangle correlation heatmap. Fit a simple linear regression model using lm, find the fitted values and confidence intervals for the fitted values given x, y by the model, draw the regression line using geom_line, and shade the confidence intervals using geom_ribbon to get the linear correlation heat map. Finally, the related genes of the key genes were screened by Spearman correlation analysis for further analysis.

### 4.5. Enrichment Analysis and Crossover Analysis

For the enrichment analysis of significant genes, we utilized the enrichGO and enrichKEGG functions from the R package clusterProfiler. The enrichGO function was employed for GO functional annotation, while the enrichKEGG function was used for KEGG pathway enrichment analysis. A significance threshold of *p* < 0.05 was applied to determine enriched terms. The GSEA (version 3.0) software was employed to conduct an analysis of the gene sets corresponding to the selected nine pathways in both the AD and SCLC environments. The c5.all.v7.2.symbols.gmt dataset was utilized, and the GSEA was carried out in accordance with the default parameter settings. The number of randompermutations was set at 1000. With the screening criteria of false discovery rate (FDR) < 0.25 and *p* < 0.05, all the nine pathways conformed to the criteria and were of significance.

### 4.6. Animal Model Groups and Drug Administry

The wild-type mice (C57BL/6J) used in this study were all males, purchased from Shanghai Southern Model Bio-Technology Co. Ltd. (Shanghai, China), and were raised until 24 weeks of age for group administration, with a total of 12 wild-type mice. The mice were randomly assigned into two groups: the WT Con group (injected with NC-agomir every other day for five consecutive times) and the WT+miR-375-3p group (injected with miR-375-3p-agomir every other day for five consecutive times). The miR-375-3p-agomir (genepharma) solution and NC-agomir (genepharma) used in this study were purchased from Suzhou Jima Biotechnology Co. (Suzhou, China). The solutions were all configured with an injection volume of 2 nmol/kg of sterile DEPC water and injected via tail vein.

### 4.7. Subcutaneous Tumor Formation Experiment in Nude Mice

Male BALB/c nude mice (4 weeks old, weighing 20–25 g, *n* = 4) utilized for the subcutaneous tumorigenesis experiments were procured from Vital River Laboratories (Beijing, China). The animals were housed in specific pathogen-free (SPF) grade animal facilities maintained at an ambient temperature of 20–26 °C and a relative humidity of 40–70%. Mice were provided with food and water ad libitum. Following a one-week acclimatization period, the experiments commenced. Cells in the logarithmic growth phase were harvested, digested, and resuspended to a concentration of 1 × 10^7^ cells/mL. The cell suspension was maintained at 4 °C using an ice bath. Using a 1 mL disposable syringe, 0.2 mL of the suspension, containing 2 × 10^6^ cells, was aspirated for each nude mouse. The cells were injected into the scapular region of the back of each nude mouse. Tumor growth was monitored three times per week. Approximately 34 days later, the nude mice developed tumors with consistent volume and regular morphology. Tumor volume was calculated using the formula: tumor volume = long diameter × (short diameter)^2^ × 0.5. In accordance with the experimental protocol, the excised tumor specimens were either stored in liquid nitrogen following lyophilization or fixed in paraformaldehyde for subsequent immunohistochemical analysis.

### 4.8. The Measurement of miR-375-3p Expression Level

Add 1 mL of TransZol Up (Transgen bio, Cat No. ER501-01-V2) (Beijing, China) to each group of cells (4 × 10^6^ cells) and mouse brain tissue (200 µL/100 mg), let stand for 5 min, add 200 µL of chloroform, shake vigorously for 30 s, incubate at room temperature for 3 min at 4 °C, centrifugate at 10,000× *g* for 15 min, collect the intermediate layer and add it to the column. min, collect the intermediate layer and add an equal volume of anhydrous ethanol, mix well and add to the centrifugal column, centrifuge at 12,000× *g* for 30 s, discard the effluent and then use the washing solution to repeat the wash two times, and then add 50 µL of RNase-free water to elute the RNA. After obtaining Total RNA, it was used as a template, and then used as a template for the synthesis of RNA using the ABScript miRNA First Strand Synthesis Kit (ABclonal, Wuhan, China) (via the Tail A) (RK30170) to synthesize the cDNA strand of miR-375-3p. In this experiment, Anchored Oligo(dT)18 was used as the primer, and the components were added and then incubated at 42 °C for 15 min, and heated at 85 °C for 5 s. The cDNA strand of miR-375-3p was synthesized using the BlasTaq^TM^ 2XqPCRMasterMix (Abmbio, Cat No.G891) (Boston, MA, USA). Data were collected and analyzed using an Applied Biosystems 7500 Real-Time PCR System (Thermo Fisher Scientific, Waltham, MA, USA). Three replicates were analyzed for each sample. Actin cDNA was normalized by the 2^−ΔΔCT^ method.

### 4.9. Total Protein Extraction and Western Blotting of Brain Tissue Samples

Cells of each group were collected, protein lysate was added, and the lysate was shaken slightly to make the lysate touch the cells completely, and the cells were processed by cell scraping after 5 min, and incubated on ice for 20 min; brain tissues of each group were collected, and the parietal lobe and frontotemporal lobe were cut out after freeze-drying in liquid nitrogen, and RIPA lysate (Meilunbio, Cat No. MA0152) (Dalian, China) was added, and the tissue samples were incubated on ice for 20 min. Tissue samples were placed on ice and broken using an ultrasonic crusher with a power setting of 100 W. The total ultrasound time for cells was controlled within 25 s, with a total interval of 30 s. The total ultrasound time for brain tissues was controlled within 40 s, with a total interval of 60 s. The brain tissues were centrifuged at 4 °C, 3000 rpm for 5 min after ultrasonication, and the supernatant was the total protein extract; brain tissues could be centrifuged in multiple ultrasound cycles in order to prevent incomplete fragmentation. The brain tissue could be centrifuged several times to prevent incomplete fragmentation. The total protein extracts were collected from each group of cells and mice, and the concentration of the total protein extracts was determined using the BCA Protein Concentration Assay Kit (Meilunbio, Cat No. MA0186). A 4× sampling buffer was added proportionally and placed in a boiling water bath for 8 min, and the processed cell samples were stored at −20 °C, and the tissue samples were stored at −80 °C. After SDS-PAGE was performed, the proteins were transferred to PVDF membrane (100 mA, 2h, 4 °C). The membranes were incubated with 5% skimmed milk for 1 h at room temperature and then incubated with primary antibodies at 4 °C overnight. After washing the membrane, it was incubated with enzyme-labelled secondary antibody at room temperature for 1 h. Finally, enhanced chemiluminescence (ECL) assay was performed and the results were analyzed using ImageJ software (Image-pro plus 6.0).

### 4.10. Tunel Assay

Apoptotic cells were identified via TUNEL staining employing a one-step TUNEL apoptosis detection kit (Meilunbio, Cat No. MA0223), with nuclei counterstained using DAPI. We captured images using a fluorescence microscope (DMI400B, Leica, Wetzlar, Germany).

### 4.11. Statistical Analysis

All the real-time PCR, Western blot, and TUNEL detections in this research were analyzed with three or more independent samples. When conducting the experimental groupings, we guaranteed the independence and randomness of the experiments. For all the mouse and cell studies, two-tailed unpaired *t*-tests or ANOVA for comparing more than two groups were employed. Statistical analyses were conducted using GraphPad Prism 8 software. All data points were utilized for statistical analysis. Data are presented as ±SEM, and *p* < 0.05 was regarded as statistically significant.

## 5. Conclusions

MiR-375-3p exerts a protective effect on neurons by up-regulating associated risk genes and promoting microglial activation in the AD environment. In the tumor microenvironment, miR-375-3p can decrease the incidence of tumorigenesis through its involvement in both cell-autonomous and non-autonomous mechanisms, matrix reprogramming, induction of apoptosis, and inhibition of tumor invasion and metastasis. Our study investigated the differential expression of miR-375-3p and associated pathways in AD and SCLC environments. Moreover, miR-375-3p plays a pivotal role in promoting cell proliferation and protecting cells from damage by precisely modulating the binding patterns of related genes to specific target genes, with significant variations observed across different biological contexts. Consequently, targeted regulation of miR-375-3p and its associated signaling pathways holds considerable promise for monitoring the progression of AD and SCLC, as well as guiding clinical interventions.

## Figures and Tables

**Figure 1 ijms-26-03666-f001:**
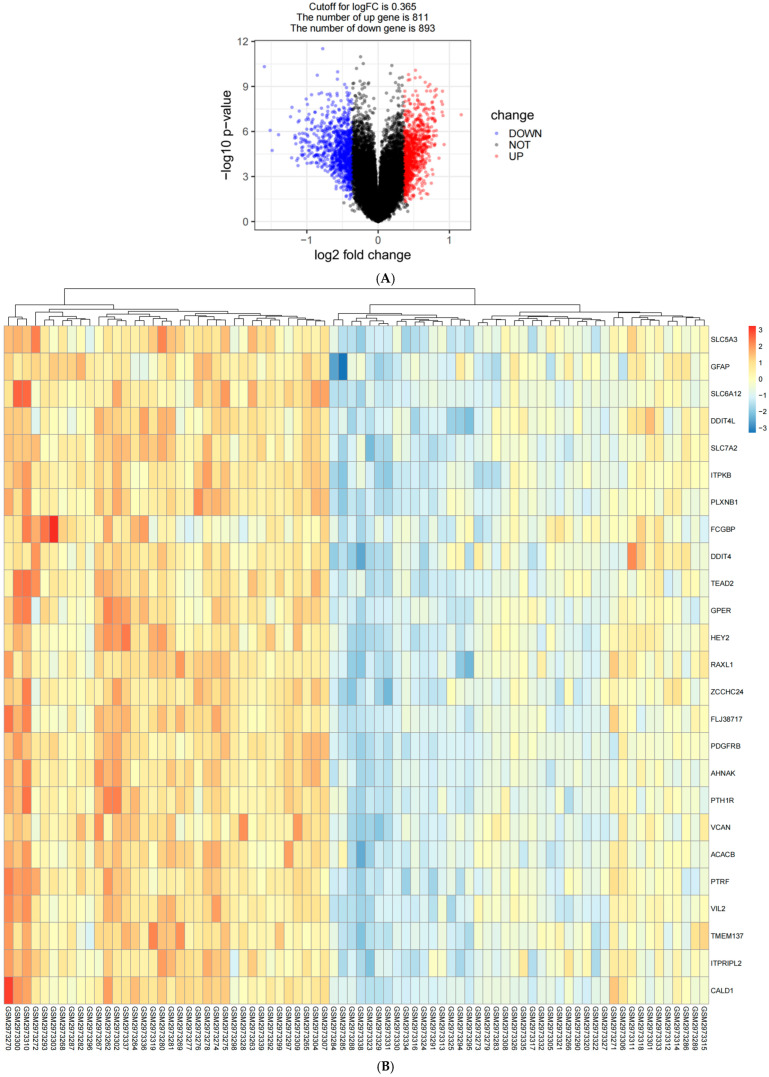
(**A**) Volcano plot of differentially expressed genes in AD. Each point in the figure represents a gene, with up-regulated genes in red, down-regulated genes in blue, and virtually no change in gene expression in black. (**B**) Heat map of top 25 AD differentially expressed genes. In the figure, the red area represents the high expression of the corresponding gene in this sample, and the blue area represents the low expression of the corresponding gene in this sample; the darker the two colors, the higher or lower the expression of the corresponding gene.

**Figure 2 ijms-26-03666-f002:**
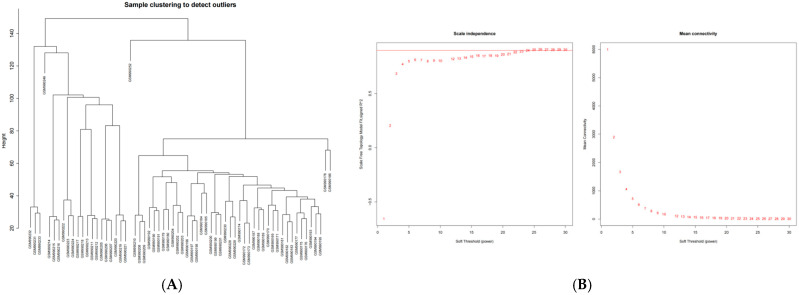
(**A**) Cluster tree of SCLC samples. The similarity of samples within each cluster was high. (**B**) Determination of the soft threshold. (**C**) Clustering trend plot, where different colors denote gene co-expression modules within the gene tree. (**D**) Heat map of the correlation between each module and SCLC traits. The plots show the relationships between modules and traits, with correlations and *p*-values shown for each cell. (**E**) Scatter plot depicting module correlations, specifically highlighting the relationship between MM and GS in the turquoise, green, and brown modules.

**Figure 3 ijms-26-03666-f003:**
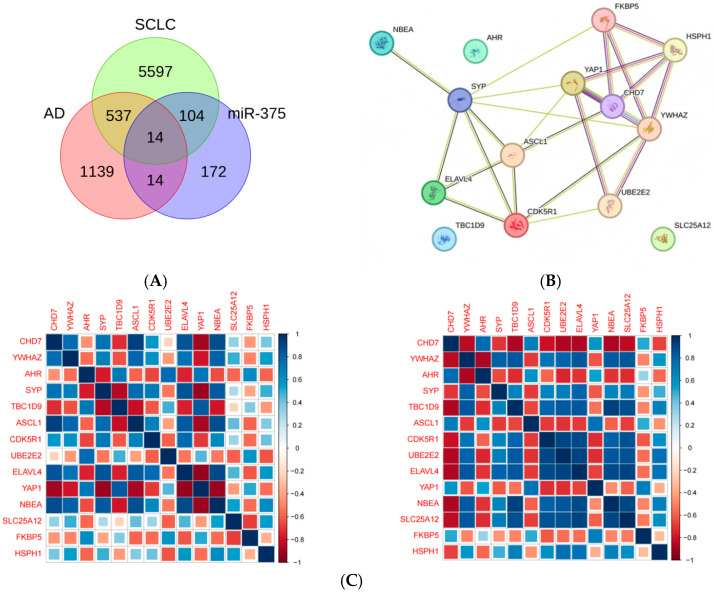
(**A**) Venn diagram illustrating the intersection of differentially expressed genes in AD, module genes in SCLC, and target genes of miR-375-3p. (**B**) PPI network, where each node represents a unique protein. Isoforms derived from the same gene are consolidated into a single node. The gene symbol is denoted by the letter on each node, and the edges between nodes represent interactions between the corresponding proteins. (**C**) The correlation heatmap illustrates the intersecting target genes in AD and SCLC. The x-axis and y-axis represent the intersecting genes, while the squares within the heatmap denote the correlation between the corresponding genes on the axes. Blue signifies a positive correlation, whereas red signifies a negative correlation. The intensity of the color reflects the strength of the correlation between the two genes on the respective axes. (**D**) Triangular correlation heatmap illustrating the intersection of target genes in AD and SCLC. The color intensity and sector area reflect the strength of correlations, where red signifies positive correlations and blue signifies negative correlations. A darker color and larger sector area denote a stronger correlation between the two genes. Asterisks indicate statistical significance, with more asterisks representing lower *p* values and higher significance. (**E**) Linear correlation plot, where each point represents an individual sample.

**Figure 4 ijms-26-03666-f004:**
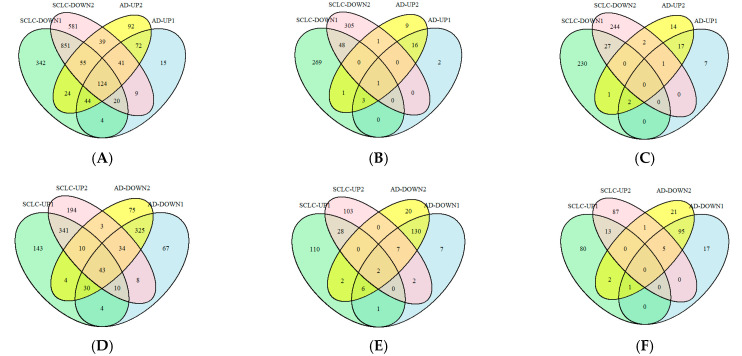
(**A**–**F**) GO analysis Venn diagram. (**A**–**C**) The pathways of ASCL1, CHD7, and their related genes up-regulated in AD and down-regulated in SCLC were intersected, with 1 representing CHD7 and 2 representing ASCL1. (**A**) In terms of BP, there are 124 intersections. (**B**) CC has one intersection. (**C**) For MF, there are zero intersections. (**D**–**F**) The intersection of pathways in which ASCL1, CHD7, and their related genes were down-regulated in AD and up-regulated in SCLC was taken, with 1 representing CHD7 and 2 representing ASCL1. (**D**) In terms of BP, there were 43 intersections. (**E**) In terms of CC, there are two intersections. (**F**) In terms of MF, there are zero intersections. (**G**,**H**) KEGG enrichment analysis Venn diagram. (**G**) Intersection of pathways of ASCL1 and its related genes up-regulated/down-regulated in AD and down-regulated/up-regulated in SCLC. (**H**) Intersection of pathways of CHD7 and its related genes up-regulated/downregulated in AD and down-regulated/up-regulated in SCLC. (**I**,**J**) GSEA and (**I**,**J**) represent the AD environment and SCLC environment, respectively. The ordinate enrichment score (ES) represents the overexpression degree of the gene set to be studied in the existing gene set in the database. The abscissa represents a gene in the gene set to be studied. The lower gray area is the control using the existing gene sets in the database as the sorting template. Each vertical bar in the middle part is a gene to be studied, the color corresponds to the upper curve, and the higher the differential expression, the higher the ranking. The genes to be studied were concentrated in the first half of the control, and the corresponding pathway was up-regulated. Conversely, the genes to be studied are concentrated in the second half of the control, which corresponds to the down-regulation of the pathway.

**Figure 5 ijms-26-03666-f005:**
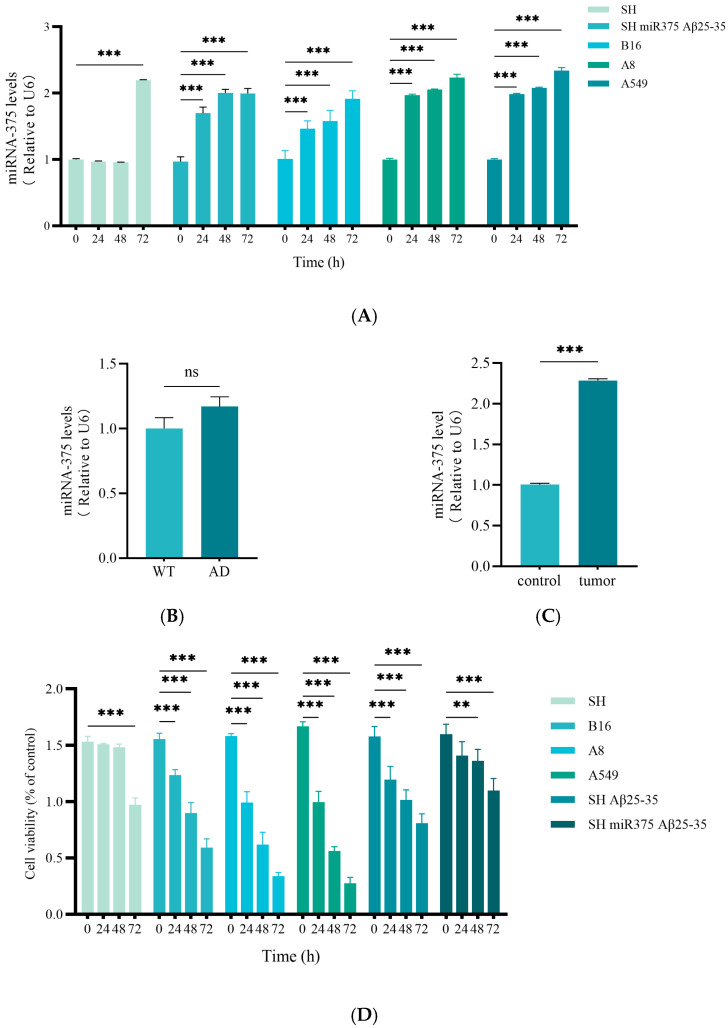
In this study, the following cell lines were utilized: human neuroblastoma SH-SY5Y cells, human bronchial epithelial 16HBE (hereinafter referred to as B16) cells, human laryngeal squamous cell carcinoma AMC-HN-8 (hereinafter referred to as A8) cells, and human NSCLC A549 cells. In our experiments, we chose the SHSY5Y cell line and A549 cell line, which corresponded to the BioSignature predictions, to correspond to the AD and SCLC disease environments, respectively. MiR-375-3p was overexpressed differently in different cell lines, which reflected the heterogeneity of its expression level and role in different disease environments. (**A**) SH-SY5Y, B16, A8, and A549 cells were seeded into culture plates at a density of 50–60%. On the following day, each cell line was transfected with either a control vector or miR-375-3p. Following a 24 h transfection period, the cells were exposed to Aβ_25-35_ for durations of 24, 48, and 72 h. Subsequently, samples were collected and the expression levels of miR-375-3p were quantified using qPCR. (**B**) The qPCR assay was employed to evaluate the expression levels of miR-375-3p in the hippocampus of both wild-type and AD mice (8 months old). The results indicated no significant difference (ns). (**C**) The qPCR assay was utilized to assess the expression levels of miR-375-3p in tumor tissues derived from squamous cell carcinoma and adjacent non-tumor tissues. (**D**) Following the treatment protocol outlined in (**A**), cells were harvested at 24 h, 48 h, and 72 h post-culture, and cell viability was measured using the CCK-8 assay. All the above experiments were repeated three times (*n* = 3), and the data were expressed as mean ± SD. One-way analysis of variance (ANOVA) and Bonferroni multiple comparison test were used for comparison. ** *p*  <  0.01, *** *p*  <  0.001, ns: not significant.

**Figure 6 ijms-26-03666-f006:**
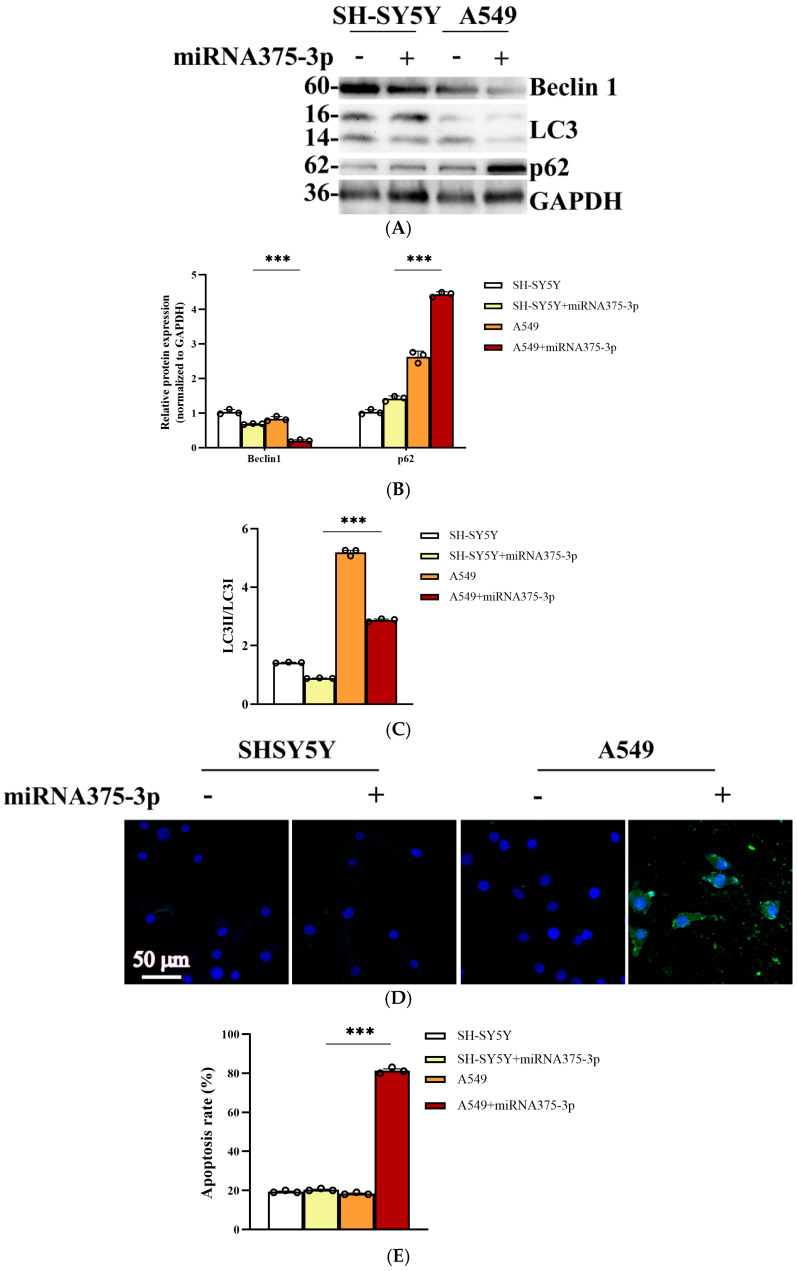
Effects of miR-375-3p on autophagy and apoptosis in different cells (**A**) Results of Western blot experiments. Different effects of miR-375-3p on autophagy markers in SH-SY5Y and A549 cells. MiR-375-3p-mimic was transfected into SH-SY5Y or A549 cells, and samples were collected 24 h later. The contents of Beclin1, LC3 and p62 were detected, and glyceraldehyde-3-phosphate dehydrogenase (GAPDH) was used as an internal control. (**B**) The gray values of Beclin1 and p62 proteins in different cell lines treated with miR-375-3p were compared with the gray values of the corresponding internal reference protein GAPDH. Finally, the ratios were normalized and statistical graphs were plotted. (**C**) Using GAPDH as the internal reference, we compared the gray value of LC3BII with that of LC3B in different cell lines after treatment with miR-375-3p. After normalizing the ratios, statistical analysis was performed. (**D**) Results of TUNEL staining analysis. Effects of miR-375-3p on apoptosis in SH-SY5Y and A549 cells. The samples were processed 24 h after cell transfection, as described in (**A**), and TUNEL assay was performed. The nuclei of apoptotic cells are shown by fluorescence, and positive apoptotic nuclei are green. The scale bar was 50 µm. (**E**) The fluorescence intensity of cells in each group was counted using Image-Pro Plus 6.0, and the cell apoptosis rate was calculated. All the above experiments were repeated three times (*n* = 3), and the data were expressed as mean ± SD. One-way analysis of variance (ANOVA) and Bonferroni multiple comparison test were used for comparison. *** *p*  <  0.001, ns: not significant.

**Figure 7 ijms-26-03666-f007:**
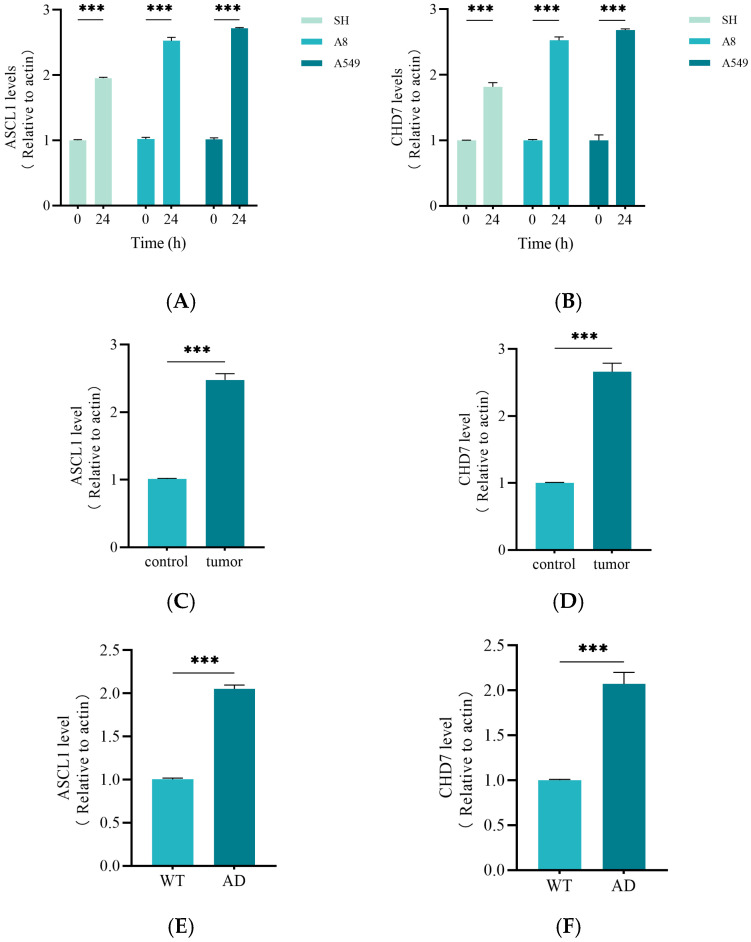
The histograms of qRT-PCR. The mRNA levels of ASCL1 and CHD7 in different cell and tissue samples were detected. (**A**,**B**) MiR-375-3p was transfected into the SH-SY5Y, A8 and A549 cell lines, respectively. After transfection for 24 h, the mRNA levels of ASCL1 and CHD7 were detected by qPCR. (**C**,**D**) Expression levels of ASCL1 and CHD7 in normal lung cells and adjacent tissues were analyzed. (**E**,**F**) ASCL1 and CHD7 expression levels were compared in hippocampal samples from the brains of wild-type mice and APP/PS1 mice. All the above experiments were repeated three times (*n* = 3), and the data were expressed as mean ± SD. One-way analysis of variance (ANOVA) and Bonferroni multiple comparison test were used for comparison. *** *p*  <  0.001, ns: not significant.

## Data Availability

All other relevant data supporting the key findings of this study are available within this article.

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
