# Peer review of "Multi-Omics Integration Analysis Revealed That miR-375-3p Is a Two-Sided Factor Regulating the Development and TUMORIGENESIS of Alzheimer’s Disease"

_ijms, 2025, doi:10.3390/ijms26083666_

Round 1

Reviewer 1 Report

Comments and Suggestions for Authors

In the manuscript entitled “Multi-omics integration analysis revealed that miR-375-3p is a two-sided factor regulating the development and tumorigenesis of Alzheimer's disease”, Bao et al aimed to study the contribution of miR-375-3p in two distinct diseases, AD and SCLC. They performed a bioinformatic analysis to identify genes that could be targeted by miR-375-3p in AD and SCLC context. They also performed cellular and mouse experiments to study the expression of mir-375-3p in cells and mouse models of AD and tumorigenesis. The authors detected a set of common potential target genes of mir-375-3p in AD and SCLC. In cellular experiments, mir-375-3p seemed to link with different potential effects in AD and SCLC via autophagy and tumor microenvironment, respectively. The authors also showed that CHD7 and ASCL1, two down-stream target genes of mir-375-3p are upregulated in AD and SCLC. The findings from this study are potential interesting, which provides some insights the contribution of mir-375-3p in two complex diseases. However the results, conclusion and discussion could be presented in a clearer way. Some of conclusions seems to be overstated. Specific comments are included below.

  1. Figures should be presented in a better way with clearly labels indicating the source data, animal study, number of replicates.
  2. Figure 6A and conclusion on autophagy and tumor environment need to be supported by quantification instead of a signal western blot
  3. It is unclear why SH-SY5Y and A549 are good models of AD and tumor environment. The conclusion from cellular experiments should reflect the nature of the models.
  4. It is unclear why mir-375-3p levels are upregulated in AD while down regulated in SCLC, but both downstream target genes are regulated in cellular models
  5. The link and flow of points throughout the discussion section can be improved when linking back to the results from the study

Author Response

  1. Figures should be presented in a better way with clearly labels indicating the source data,

animal study, number of replicates.

We sincerely appreciate your valuable comments. We have carefully examined the Materials and Methods section of the article. Revisions and improvements have been made to aspects such as the source of experimental data, animal studies, and the number of repetitions in Sections 4.6, 4.8, and 4.9 of the article.

  1. Figure 6A and conclusion on autophagy and tumor environment need to be supported by quantification instead of a signal western blot.

Thank you for your constructive comments on our article. We statistically analyze the relative expression results of proteins and the TUNEL-positive cells in different cell lines treated with miR-375-3p. We have newly added Figures B, C, and E to Figure 6 to support our conclusion that cellular autophagy is related to the tumor environment with quantitative statistical results. We have also made adjustments in the expression in the article 2.6. The revisions are as follows.“The fluorescence intensity of the staining results of each group was analysed using Image-pro plus 6.0, and the results are shown in Figure 6E. There was no significant change in the apoptosis rate after the addition of miR-375-3p in the SH-SY5Y group, and the apoptosis rate increased significantly after the addition of miR-375-3p in the A549 group, which was about 4 times of the control. The above results indicated that miR-375-3p overexpression alone had no significant effect on apoptosis in neuronal cells, but could inhibit lung cancer cell survival through pro-apoptotic mechanism, which was consistent with the previous results.”

  1. It is unclear why SH-SY5Y and A549 are good models of AD and tumor environment. The conclusion from cellular experiments should reflect the nature of the models.

We sincerely appreciate your valuable suggestions. We have supplemented the relevant information in Figure 5 of the article, and the specific content is as follows: "In our experiments, we chose the SHSY5Y cell line and A549 cell line, which corresponded to the BioSignature predictions, to correspond to the AD and SCLC disease environments, respectively. miR-375-3p was overexpressed differently in different cell lines, which reflected the heterogeneity of its expression level and role in different disease environments."

SH-SY5Y cells are derived from neuroblastoma and are highly sensitive to Aβ25-35 (the core toxic fragment of Alzheimer's disease, AD). This is highly consistent with the neuronal degenerative lesions in AD patients, and SH-SY5Y cells can be used to simulate the neuroinflammatory microenvironment of AD. A549 cells are derived from human lung adenocarcinoma. They show a significant response to autophagy inhibition (such as the decrease of LC3-II and the accumulation of p62 caused by the overexpression of miR-375-3p) and apoptosis induction (such as the increase in the positive rate of TUNEL). A549 cells are suitable for studying the metabolic reprogramming and immune escape in the tumor microenvironment. This can reflect the survival strategies of tumor cells under metabolic stress and is highly consistent with the clinicopathological characteristics of solid tumors (such as lung cancer).

  1. It is unclear why mir-375-3p levels are upregulated in AD while down regulated in SCLC, but both downstream target genes are regulated in cellular models.

Thank you very much for your professional comments on our article. We have made further revisions in the article 4.1 to discuss this issue carefully. Regarding the expression of miR-375-3p in two different disease environments, Alzheimer's disease (AD) and small cell lung cancer (SCLC), we have analyzed it from the perspective of pathways in the Discussion section of the article to illustrate that the function of miRNA depends on the direction of expression change and the cellular environment. The changes are as follows.

"In AD, miR-375-3p usually inhibits pro-apoptotic or pro-inflammatory genes to exert a neuroprotective effect. The up-regulation of miR-375-3p levels in AD may be related to the activation of signaling pathways induced by inflammatory responses, oxidative stress, or β-amyloid deposition. In SCLC, miR-375-3p usually functions as a tumor suppressor gene. The down-regulation of miR-375-3p levels in SCLC will lead to increased expression of target genes (such as MYC and MCL1), promoting the proliferation and survival of tumor cells. At the same time, the function of target genes is bidirectional. In AD, the up-regulation of miR-375-3p inhibits pro-inflammatory pathways and alleviates neuronal damage. In SCLC, its down-regulation relieves the inhibition of oncogenes and promotes tumor progression. Therefore, both downstream target genes of miR-375-3p are regulated in cellular models, and the functions they exert only depend on the direction of expression change and the cellular environment."

  1. The link and flow of points throughout the discussion section can be improved when linking

back to the results from the study.

Thank you very much for your valuable suggestions. It will contribute to enhancing the professionalism and integrity of our article. Based on the overall content of the article, we have rewritten the discussion section and highlighted it in the article.

Reviewer 2 Report

Comments and Suggestions for Authors

Comments attached.

Author Response

Major Comments

  1. Figure 1. Please arrange the figures properly and increase the font size to increase readability. Cannot read anything even after magnifying the images.

Thank you for pointing this out. We have replaced all images in the article with the original images to ensure clarity. Additionally, we have created a PDF of the group images and legends, which has been added as an attachment. We have also combined all original images into a single compressed file, which is included as an attachment.

  1. Line 188, the authors introduced AD first and SCLC later, but in the results 3.1 is about SCLC. I suggest rearranging the introduction.

We sincerely thank you for your valuable comments. We carefully checked the literature, revised the logical relationship between AD and SCLC, introduced AD first and then SCLC in the results section, and checked the whole article to modify the problems with similar logical order.

  1. Result 3.1, please start with the purpose of the experiment.

As you suggested, we have improved the language of the results section and summarized the purpose of the experiment for clarity at the beginning of each paragraph in the results section.

  1. Line 205, please explain what MM is and its significance.

Thank you for pointing this out. We agree and have added a further explanation of MM in Section 2.2 of the article. The revised content is as follows. "Module membership (MM) refers to the affiliation or degree of participation of a gene within a module. A network is constructed by calculating the expression correlation between genes, and genes with similar expression patterns are clustered into modules. In the research on disease-related gene networks, it has been found that certain genes have high MM expression in disease-related modules. This indicates that these genes may be key driver genes of the disease or potential therapeutic targets."

  1. Result 3.2, please explain the purpose of the experiment and why you chose the temporal gyrus?

We are grateful for your professional comments on our article and have made further changes to discuss this issue in detail in Section 2.1 of the article. The changes are as follows. "We chose the middle temporal gyrus as a target area for studying the molecular mechanisms of AD for the following reasons. First, the middle temporal gyrus is a routine sampling site in neuropathological studies, and a large amount of transcriptomic data of the middle temporal gyrus tissues have been accumulated in the clinical sample bank; Second, the temporal lobe is one of the earliest and most significantly affected brain regions in AD, and the middle temporal gyrus is an important part of the temporal lobe."

  1. Line 246, the heading is not understandable. What is the meaning of intersect here?

We appreciate the reviewer's suggestion. In response, we have rephrased the title 2.3 according to the reviewer's advice and revised it to "Screening for miR-375-3p Target Genes Based on AD Differential Genes and SCLC Module Genes."

  1. Line 248, what is psRNATarget?

Thank you for highlighting this point. We agree and have added a further explanation of psRNATarget in the section 3.1 of the article. The revised content is as follows: "psRNATarget is an online tool that comprehensively searches for potential miRNA targets across the entire genome based on small RNA and mRNA sequence information. Once miRNAs that may interact with a specific mRNA are identified, the focus can be narrowed to the regulatory effects of these miRNAs on the mRNA, thereby reducing the scope of the research."

  1. Figure 4, Why authors are comparing gene sets between AD and SCLC. Is there a Connection between AD and SCLC? I do not understand the logic for this comparison. Explain I and J - The legends are not readable.

Thank you for kindly reminding us to clarify this point, and we have further revised the article to make the answer to this question clearer. Our previous study suggested that miR-375-3p may have functional heterogeneity in different pathological microenvironments. In this study, two disease models with significant differences in pathological characteristics between AD and SCLC were selected to systematically verify this phenomenon and elucidate its molecular mechanism. The intersection of AD differential genes, SCLC module genes and miR-375-3p target genes was used to construct a cross-disease co-regulation network to find the bidirectional regulation mode of core regulatory genes in different diseases. No comparisons were made between AD and SCLC Settings.

At the same time, we have adjusted the expression of legend I-j in Figure 4 and modified it as follows: "(I and J) represent AD environment and SCLC environment respectively. The ordinate ES represents the overexpression degree of the gene set to be studied in the existing gene set in the database. The abscissa represents a gene in the gene set to be studied. The lower gray area is the control using the existing gene sets in the database as the sorting template. Each vertical bar in the middle part is a gene to be studied, the color corresponds to the upper curve, and the higher the differential expression, the higher the ranking. The genes to be studied were concentrated in the first half of the control, and the corresponding pathway was up-regulated. Conversely, the genes to be studied are concentrated in the second half of the control, which corresponds to the down-regulation of the pathway."

  1. Results 3.5. Start with purpose of the experiment for clarity. What are A8 and A549 cells? Why were these cell lines used? The authors used multiple cell lines with some numbers that are not understandable. Explain the experiment properly, what type of cell lines were used, and why these were used.

As you suggested, we have improved the language of the results section and summarized the purpose of the experiment for clarity at the beginning of each paragraph in the results section.

We sincerely appreciate your valuable suggestions. We have supplemented the relevant information in Figure 5 of the article, and the specific content is as follows: "In our experiments, we chose the SHSY5Y cell line and A549 cell line, which corresponded to the BioSignature predictions, to correspond to the AD and SCLC disease environments, respectively. miR-375-3p was overexpressed differently in different cell lines, which reflected the heterogeneity of its expression level and role in different disease environments."

The A549 cells are derived from human lung adenocarcinoma and have a significant response to autophagy inhibition (such as the decrease of LC3-II and the accumulation of p62 caused by the over expression of miR-375-3p) and apoptosis induction (such as the increase of the TUNEL positive rate). They are suitable for studying the metabolic reprogramming and immune escape in the tumor microenvironment. This can reflect the survival strategies of tumor cells under metabolic stress and is highly consistent with the clinicopathological characteristics of solid tumors (such as lung cancer). A8 (AMC-HN-8) is a cell line derived from human laryngeal squamous cell carcinoma, and it can serve as a good control group with normal laryngeal cancer cells. A8 cells retain the molecular characteristics related to laryngeal cancer (such as EGFR amplification and activation of the PI3K/Akt pathway), and are often used to simulate the local invasion and lymph node metastasis behavior of laryngeal cancer. They are suitable for studying the specific mechanisms of squamous cell carcinoma.

  1. Line 429, Where is the data showing association with Autophagy/apoptosis?

Thank you for your correction. In our article, it is stated that "bioinformatics analysis shows that autophagy and its key pathways are significantly correlated with the sample data." Among them, the pathway related to autophagy is "gene mutation leads to the abnormal accumulation of β-amyloid (Aβ) and the protein degradation mediated by the 26S proteasome," which is elaborated in detail in the 2.4 Results section and is reflected in Figure 4(I)(J). Autophagy and the proteasome pathway are the two major main pathways for intracellular protein degradation. Although this pathway mainly expounds on the proteasome pathway, in situations such as neurodegenerative diseases, the autophagic process may also be involved in the clearance of abnormal proteins.

To more clearly present the relationship between miR-375-3p and autophagy, we have added two new figures (Figure 4 I-J) in section 2.4 of the article to explore the differential regulation of the PI3K-Akt signaling pathway in different disease environments. The PI3K-Akt signaling pathway is directly related to autophagy. Therefore, we have supplemented the introduction of this pathway in the Discussion 3.3 section: "Akt (protein kinase B) is crucial in the regulation of autophagy and is one of the main activators of the mammalian target of rapamycin (mTOR). As a key regulatory protein, mTOR plays a significant role in the initiation of autophagy. When Akt activates mTOR, mTOR will inhibit autophagy; in addition, Akt can directly phosphorylate and inhibit a variety of autophagy-related proteins such as TSC2 and FoxO family transcription factors, and these proteins play an important role in autophagy induction."

  1. Line 431, what was the level of over expression of miR-375-3p? Please add a graph showing the difference in the level in your experiments.

We performed the experiment again according to your suggestion and obtained the qRT-PCR results concerning the expression level of miR-375-3p. We described and analyzed the results in Section 2.5 of the article, and replaced figure5(A). The changes are as follows. “In the SH-SY5Y cell line, we added Aβ25-35 to establish an AD disease model. The qRT-PCR results showed that the presence of Aβ25-35 caused an early upregulation of miR-375-3p expression, with a significant increase observed at 24 and 48 hours. miR-375-3p was significantly overexpressed in A8 and A549 cell lines 24 hours after transfection, and this high expression persisted for 72 hours. At 24 hours after transfection, the expression of miR-375-3p in B16 epithelial cells showed an upward trend, but the growth rate was slower than that in squamous cell carcinoma cells, and reached a significant level at 48-72 hours. In Section 3.4, we obtained that the target genes of miR-375-3p are involved in the regulation of PI3K-Akt signaling pathway, which is closely related to the induction of apoptosis and autophagy damage. This experiment proved that the expression level of miR-375-3p is different in different disease Settings. This is related to the differential regulation of miR-375-3p in different disease Settings. (Figure 5A).”

  1. Line 498, vague sentence saying protective effect on cells, but no data, what protective functions?

Thanks for your important suggestion, we re-performed CCK-8 experiment and obtained the results of cell damage and protection. We described and analyzed the results in Section 2.5 of the article, and replaced figure5(D). The changes are as follows. “The results of CCK-8 experiments showed that overexpression of miR-375-3p significantly reduced the survival of B16, A8 and A549 cell lines compared with SHSY5Y cells. We hypothesized that miR-375-3p could induce cell injury and autophagy and apoptosis. In SH-SY5Y cell line, Aβ25-35 was added to establish AD disease model. The addition of Aβ25-35 could cause cell damage, and the degree of damage increased with the increase of time (24 h, 48 h and 72 h). However, the number of surviving cell lines was increased after re-addition of miR-375-3p overexpression compared with SHSY5Y Aβ25-35. This indicated that miR-375-3p overexpression alleviated the cytotoxicity induced by Aβ25-35, thereby protecting the cells. CCK8 results further demonstrate that miR-375-3p plays a dual role in glioma cells and tumor cells.(Figure 5D)”

  1. Figure 6A. looks like some western blots are inverted vertically, especially p62 and LC3.

We greatly appreciate your professional comments on our article, and as you are concerned, we will explain the answer to this question for you in detail. During autophagy, LC3-II is a marker of autophagosome formation, and its level is usually positively correlated with autophagic activity. However, p62 acts as an adaptor protein for autophagy substrates, and its accumulation reflects the blockage of autophagic flow. In the SH-SY5Y cell line, miR-375-3p over expression only slightly inhibited Beclin1, but did not significantly affect LC3-II and p62 levels, indicating that the autophagic flow was generally balanced. In A549 cell line, miR-375-3p significantly inhibited the expression of Beclin1 and LC3-II, but accumulated a large amount of p62, suggesting that both autophagy initiation and autophagosome formation were inhibited. miR-375-3p promotes immune escape through "autophagy inhibitor-p62 accumulation" in SCLC, while it protects neurons by maintaining autophagy homeostasis in AD. This highlights the environmental dependence of miR-375-3p function and the complexity of therapeutic targets.

Minor Comments

  1. Line 65, change “In this experiment” to “In this study”.

We sincerely appreciate the reviewers for their careful reading. According to the reviewers' suggestions, we have changed "In this experiment" to "In this study".

  1. Line 192, What is WGCNA?

Thank you for bringing this to our attention. We agree and have included a further explanation of WGCNA in section 4.2 of the article. The revised content is as follows: "The Weighted Gene Co-expression Network Analysis (WGCNA) is a commonly used method in bioinformatics. It calculates the expression similarity between genes based on gene expression data. By employing algorithms such as hierarchical clustering, genes with similar expression patterns are grouped into different modules. In the study of cancers and other diseases, WGCNA can identify gene modules and key genes associated with the onset and progression of diseases, providing targets for disease diagnosis and treatment."

  1. Figure 4, Too many acronyms in the legend makes it tough to follow. Please expand them.

Thank you for pointing this out. We have written the full names of the abbreviations for the first time in Figure 4. At the same time, we attached the full text abbreviation table in the attachment.

  1. The authors concentrate on ASCL1 and CHD7. Please explain what they are and their significance.

We sincerely appreciate your suggestions, which will help improve the quality of our paper. We have made further revisions to the Discussion 3.2 section of the article to carefully discuss ASCL1 and CHD7. The specific revised content is as follows. "CHD7 and ASCL1 are two key genes closely related to miR-375-3p discovered in this study. As a chromatin remodeling factor, CHD7 plays an important role in neural development and gene expression regulation." "As a transcription factor, ASCL1 plays a key role in neuronal differentiation and survival."

In section 2.3, we took the target genes of miR-375-3p and intersected them with the differential genes of AD and the module genes of SCLC, and obtained 14 key genes. Among them, the two key genes, CHD7 and ASCL1, have the highest connectivity in the protein-protein interaction network (Figure 3B). Both of them have a strong correlation with AD and SCLC, and the p-values are both < 0.05. Figure 3E shows that ASCL1 and CHD7 are significantly positively correlated in AD (R = 0.69), suggesting that the two may cooperatively regulate neurodegeneration; at the same time, these two genes are highly coordinated in SCLC (R = 0.92), suggesting that they may drive the progression of SCLC by co-regulating the neuroendocrine pathway. Both downstream target genes (ASCL1 and CHD7) of miR-375-3p are regulated in cellular models, but they play different roles in different environments. This illustrates that the function of miRNA depends on the expression changes and the disease environment.

  1. Result subheadings are vague and the name of the experiment. Please change the subheadings to what was the major discovery from the experiment.

Thank you for pointing this out. We have revised the subheadings in the results and those in the legend, and highlighted them in the article.

  1. Please cite necessary articles. Many

We sincerely thank you for your valuable comments. As you suggested, we have replaced and supplemented the necessary references and made revisions in the article.

Reviewer 3 Report

Comments and Suggestions for Authors

Summary

This study investigates the role of miR-375-3p in both Alzheimer's disease (AD) and small cell lung cancer (SCLC) using bioinformatics analysis, in vitro experiments, and animal models. The findings suggest that miR-375-3p exhibits opposing functions in these diseases—promoting apoptosis in SCLC while regulating neurodegenerative pathways in AD. The authors identified key target genes (ASCL1 and CHD7) and explored differentially expressed pathways linked to lipid metabolism, autophagy, and apoptosis. These results highlight miR-375-3p as a potential biomarker and therapeutic target for both neurodegeneration and tumorigenesis.

Questions for the Authors

Methodology Clarification:

You mention using Spearman correlation analysis to identify key target genes (ASCL1 and CHD7). However, correlation does not imply causation. Have you considered additional validation methods, such as knockdown or overexpression experiments in both disease models?

Functional Differences in Disease Contexts:

Your results indicate that miR-375-3p enhances apoptosis in SCLC but may protect neurons in AD. Could you elaborate on the underlying mechanistic differences driving this paradoxical effect, particularly regarding the role of autophagy pathways?

Potential Clinical Applications:

Given that miR-375-3p is differentially expressed in AD and SCLC, do you propose that it could serve as a biomarker for early detection? If so, what would be the feasibility of using miR-375-3p in clinical diagnostics for these diseases?

Comments on the Quality of English Language

Good

Author Response

  1. Methodology Clarification:

You mention using Spearman correlation analysis to identify key target genes(ASCL1 and CHD7). However, correlation does not imply causation. Have you considered additional validation methods, such as knockdown or over expression experiments in both disease models?

Thank you for your constructive suggestions, which have provided new insights for our research. We agree with the feasibility of the method you proposed. However, this paper focuses on the differential regulatory functions of miR-375-3p in various disease contexts, rather than the genes ASCL1 and CHD7. The selection of these two genes was to substantiate the role of miR-375-3p. miR-375-3p is located upstream of the regulatory regions of these genes. Knocking out or overexpressing ASCL1 and CHD7 does not affect the function of miR-375-3p. Therefore, manipulating these two genes in isolation cannot fully elucidate the regulatory differences of miR-375-3p across different disease settings. Your suggestion is correct and feasible, and we may conduct experiments based on it in our subsequent research. Once again, we appreciate your suggestions.

  1. Functional Differences in Disease Contexts:

Your results indicate that miR-375-3p enhances apoptosis in SCLC but may protect neurons in AD. Could you elaborate on the underlying mechanistic differences driving this paradoxical effect, particularly regarding the role of autophagy pathways?

The paradoxical effects of miR-375-3p in SCLC and AD may be attributed to the differential cellular contexts and target genes involved in each disease. In SCLC, miR-375-3p likely promotes apoptosis by down regulating anti-apoptotic genes or up regulating pro-apoptotic factors, potentially through the inhibition of autophagy-related pathways. Conversely, in AD, miR-375-3p may enhance neuronal survival by promoting autophagy, which helps clear toxic protein aggregates like Aβ and tau. The distinct roles of autophagy in cancer (where it can promote survival) and neurodegenerative diseases (where it can be protective) further explain this dual functionality.

  1. Potential Clinical Applications:

Given that miR-375-3p is differentially expressed in AD and SCLC, do you propose that it could serve as a biomarker for early detection? If so, what would be the feasibility of using miR-375-3p in clinical diagnostics for these diseases?

The differential expression of miR-375-3p in AD and SCLC suggests its potential as a disease-specific biomarker. However, its feasibility for clinical application needs to be evaluated from multiple perspectives. Firstly, miRNA detection technologies (such as qPCR or RNA-seq) are well-established and can efficiently and accurately measure the expression levels of miR-375-3p in clinical samples, which supports its technical feasibility. However, the expression of miR-375-3p may be influenced by various factors, such as age, other disease states, or therapeutic interventions. Therefore, it is necessary to combine it with other disease-specific markers to improve diagnostic accuracy and specificity. Additionally, the clinical value of miR-375-3p as a biomarker requires further validation in large cohort studies to clarify the correlation between its expression levels and disease progression, staging, or prognosis. Future work can build on the molecular mechanisms of miR-375-3p in different diseases to further explore its potential as a multi-disease biomarker and develop more precise diagnostic models through integrated multi-omics analyses (such as combining mRNA, protein, or metabolite data).

Round 2

Reviewer 2 Report

Comments and Suggestions for Authors

The authors addressed most of my comments.

Please add all the data points in the bar graph and indicate each bar's total data points/replicates. In the legend indicate what the error bars are - standard dev or standard error etc.

Author Response

Thank you very much for your constructive comments, we replaced the B, C and E images in figure 6 and changed the figure6 legend with the following modifications: ‘Data are presented as ± SEM, and p < 0.05 was regarded as statistically significant’ and ‘The fluorescence intensity of cells in each group was counted and apoptosis rate was calculated using Image-pro plus 6.0. Data are presented as ± SEM, and p < 0.05 was regarded as statistically significant.’

We also described the method in part 4.11. It said that “All the Real-time PCR, Western Blot and Tunel detections in this research were analyzed with three or more independent samples.”.